# Anti-SARS-CoV-2 Antibody Status at the Time of Hospital Admission and the Prognosis of Patients with COVID-19: A Prospective Observational Study

Ján Jurenka [1], Anna Nagyová [1], Mohammad Dababseh [1], Peter Mihalov [1], Igor Stankovič [1], Vladimír Boža [2], Marián Kravec [2], Michal Palkovič [3], Martin Čaprnda [4] and Peter Sabaka [1,*]

1   Department of Infectology and Geographical Medicine, Faculty of Medicine,
    Comenius University in Bratislava, 831 01 Bratislava, Slovakia
2   Department of Applied Informatics, Faculty of Mathematics, Physics and Informatics,
    Comenius University in Bratislava, 842 48 Bratislava, Slovakia
3   Department of Pathology, Faculty of Medicine, Comenius University in Bratislava, 811 08 Bratislava, Slovakia
4   1st Department of Internal Medicine, Faculty of Medicine, Comenius University in Bratislava,
    811 08 Bratislava, Slovakia
*   Correspondence: petersabaka@gmail.com

**Abstract:** The association between COVID-19 severity and antibody response has not been clearly determined. We aimed to assess the effects of antibody response to SARS-CoV-2 S protein at the time of hospital admission on in-hospital and longitudinal survival. Methods: A prospective observational study in naive hospitalised COVID-19 patients. The presence of anti-S SARS-CoV-2 IgM and IgG was evaluated using a lateral flow assay at the time of admission. The patients were followed up for 8–30 months to assess survival. We recruited 554 patients (330 men and 224 women). Overall, 63.0% of the patients had positive IgG or IgM anti-S SARS-CoV-2 antibodies at the time of hospital admission. In the univariate analysis, the patients with negative anti-S SARS-CoV-2 IgM and IgG antibodies were referred to the hospital sooner, had lower CRP and D-dimer concentrations, and were hospitalised longer. They were also more likely to be admitted to an intensive care unit and more often received baricitinib treatment. During their hospital stay, 8.5% of the antibody-positive and 22.3% of the antibody-negative patients died ($p = 0.0001$). The median duration of the follow-up was 21 months. During the follow-up after hospital discharge, 3.6% of antibody-positive and 9.1% of antibody-negative patients died ($p = 0.027$). In the multivariate analysis, the negative anti-S SARS-CoV-2 antibodies were associated with a higher risk of in-hospital death (OR 3.800; 95% CI 1.844–7.829; $p = 0.0001$) and with a higher risk of death during follow-up (OR 2.863; 95% CI 1.110–7.386; $p = 0.030$). These associations were independent of age, the time from symptom onset to hospital admission, CRP, D-Dimer, the number of comorbidities, disease severity at the time of hospital admission, and baricitinib therapy. Our study concludes that negative anti-S SARS-CoV-2 IgM and IgG at the time of admission are associated with higher in-hospital mortality and cause a higher risk of all-cause death during follow-up after discharge.

**Keywords:** anti-S SARS-CoV-2 antibodies; COVID-19; prognosis

## 1. Introduction

It is estimated that coronavirus disease 2019 (COVID-19) has caused almost 15 million deaths during the first two years of the pandemic [1]. Most patients with COVID-19 experience mild, self-limiting disease. However, the disease might be complicated by severe interstitial pneumonia that may result in acute hypoxemic respiratory failure and death [2]. Known predictors of disease severity and a poor outcome are the male sex, comorbidities, advanced age, obesity, elevated biomarkers of organ damage, elevated biomarkers of inflammation, lymphopenia, evidence of substantial lung involvement, and the presence of

hypoxemia [3,4]. Delayed antibody response against severe acute respiratory syndrome coronavirus 2 (SARS-CoV-2) antigens has been identified as a predictor of in-hospital mortality [5,6]. The presence of neutralising antibodies within the first weeks of the disease is also associated with earlier virus clearance and the probability of survival [7]. However, this topic remains controversial because previous studies have reported an association between high antibody titres in hospitalised patients and more severe disease [8–11]. Anti-S antibodies play a crucial role in recovering from COVID-19. Neutralising anti-S antibodies prevents the virus from entering cells, limits the extent of infection, and thus prevents the development of tissue damage in the affected organs, like lungs or myocardium [12,13]. Anti-S monoclonal antibodies administered at the early stage of the infection prevent the development of severe disease [14]. Therefore, the early development of anti-S antibodies might result in lower organ damage, less severe disease, and a better long-term prognosis. The effects of the dynamics of anti-S antibody response on the long-term prognosis of COVID-19 patients is yet to be determined. The aim of our study was to assess the effects of delayed anti-S antibody response on the risk of in-hospital death and on the risk of death during the long-term follow-up in discharged COVID-19 patients.

## 2. Materials and Methods

### 2.1. Design

We conducted a prospective observational study to determine the effects of the presence of anti-S SARS-CoV-2 antibodies at the time of admission on the prognosis of unvaccinated and naive patients hospitalised with COVID-19. The prognosis was assessed as in-hospital survival and survival after discharge during the follow-up period (Figure 1).

### 2.2. Patients

We enrolled all patients meeting the inclusion criteria admitted to the Department of Infectology and Geographical Medicine, University Hospital in Bratislava, between 1 April 2020 and 30 December 2021. The inclusion criteria were COVID-19 infection confirmed by polymerase chain reaction for SARS-CoV-2 RNA from the nasopharyngeal swab at the time of admission with a moderate, severe, or critical disease as a reason for hospital admission as defined in the National Institutes of Health (NIH) guidelines [15]. The exclusion criteria were previous vaccination with any vaccine against COVID-19 and a history of SARS-CoV-2 infection. At the time of admission, patients signed informed consent, underwent clinical evaluation and a full medical history, and anthropometric data were obtained. The disease severity was assessed using criteria according to NIH guidelines [15]. The medical history was obtained by questionnaire and from the local electronic database. Additionally, venous blood was drawn to measure the concentrations of creatinine, C-reactive protein (CRP), D-dimer, and interleukin-6 (IL-6) for a complete blood count, and to assay anti-S SARS-CoV-2 IgM and IgG antibodies. The endpoint of the study was all-cause death. The patients were followed up during hospital stay. For the discharged patients, survival was assessed using a nationwide database of the Health Care Surveillance Authority, which records the exact date of all deceased patients [16]. Survival was assessed on the same day for all patients on 17 October 2022. The follow-up duration was from 9 to 30 months (median 21 months). Regarding immunomodulatory therapy, all patients admitted due to severe COVID-19 received dexamethasone at a dose of 6 mg daily. Patients who required high-flow oxygen or mechanical ventilation were treated with baricitinib at a dose of 4 mg daily, according to the institutional guidelines. All patients who died in the hospital with COVID-19 underwent autopsy. The autopsy concluded that COVID-19 and its complications were a cause of death in all patients except four with different causes (perforation of duodenal and gastric ulcers, colorectal carcinoma, bleeding from an aneurysm of the abdominal aorta) who were removed from the analysis.

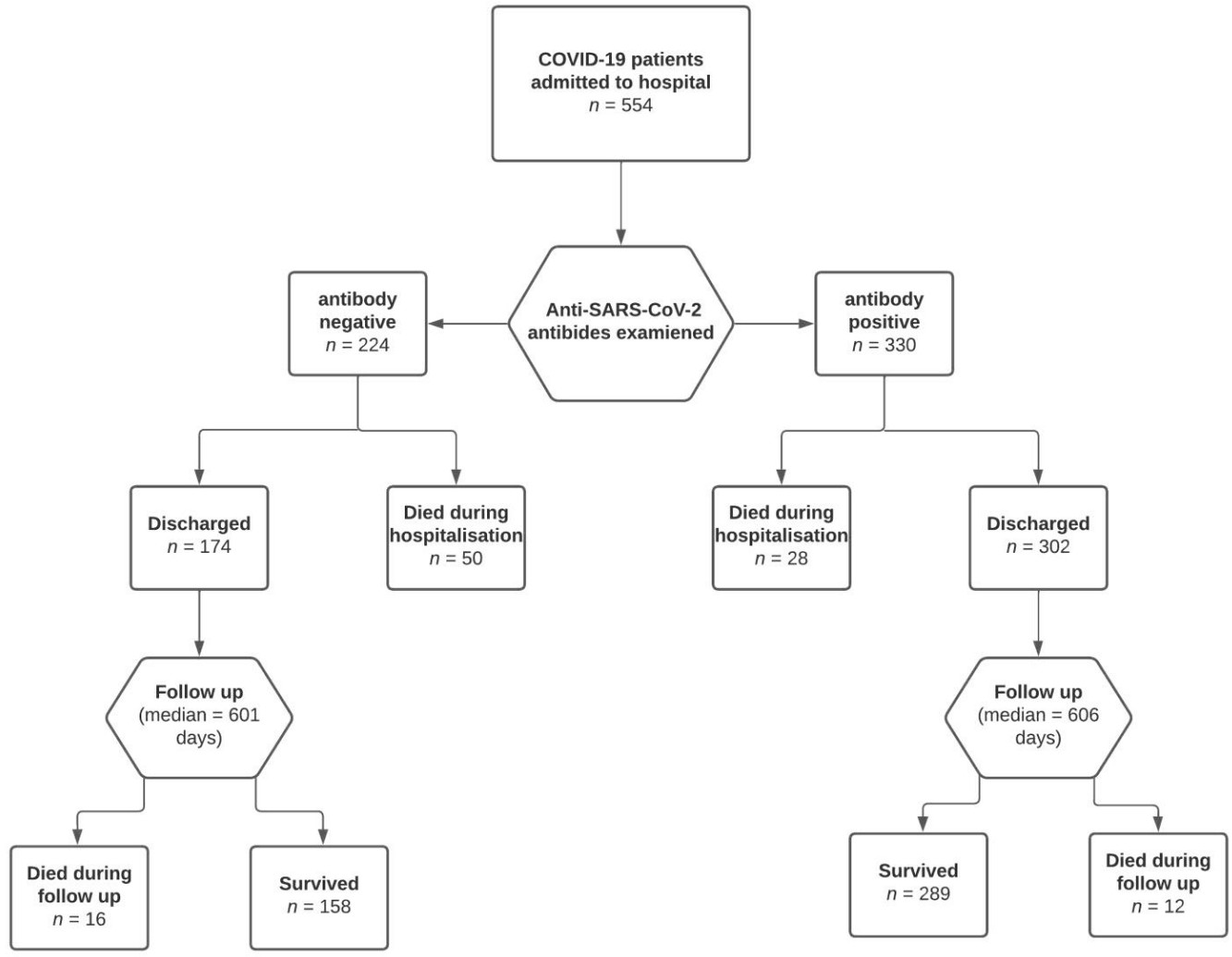

**Figure 1.** Study design.

*2.3. Biochemical Analysis*

Concentrations of creatinine were measured using spectrophotometry (Cobas Integra 400, Roche Diagnostics, Rotkreuz, Switzerland). CRP and D-dimer were measured using immunoturbidimetry (Cobas Integra 400, Roche Diagnostics). Serum IL-6 concentrations were measured using an immunoassay (Elecsys, Roche Diagnostics). The presence of anti-S SARS-CoV-2 IgG and IgM antibodies was assessed using a rapid lateral flow assay (COVID-19 IgG/IgM Rapid Test Cassette, Zhejiang Orient Gene Biotech Co., Ltd., Huzhou, Zhejiang, China). Briefly, 5 µL of serum blood was added to the test slide, followed by 80 µL of the buffer provided in the kit. The results were read after 15 min (max 20 min) by the naked eye by the same laboratory technician. Only tests in which the control line changed colour were regarded as valid. If a line was observed for IgM and/or IgG, the test was considered positive. If the patient had two or more antibody tests, only the result of the first test was included in the analysis.

*2.4. Statistical Analysis*

Quantitative variables with a normal distribution are expressed as mean ± standard deviation. Variables that are not normally distributed are expressed as medians, and the dispersion of the variables is by the 25th and 75th percentile. The distributions of the variables were assessed using Kolmogorov–Smirnov testing (Supplementary Table S1). The independent value *t*-test was used to compare the means of the variables with a normal distribution, and a Mann–Whitney U test was used to compare the medians

of the quantitative variables without normal distribution. We evaluated the distributions of the qualitative variables among the groups by using the chi-square test. We assessed the associations between in-hospital death and other variables by using a multivariate binary logistic regression model. We assessed the association of death during the follow-up after discharge with other variables using Cox regression. The presence of anti-S SARS-CoV-2 antibodies at the time of hospital admission, the duration of symptoms, the number of comorbidities, disease severity (presence of moderate, severe, or critical disease), age, baricitinib treatment, the neutrophil-to-lymphocyte ratio, body mass index, CRP, D-dimer, and IL-6 concentrations were included in the binary logistic regression and Cox regression analyses at the baseline. We adopted a forward stepwise method (probability for stepwise: entry $p < 0.05$) for binary logistic regression and Cox regression analysis to reduce the number of independent variables entering the model to reduce the probability of model overfitting. We used logistic regression instead of Cox regression to analyse the risk factors of in-hospital deaths because of the inconsistency of hospitalisation length during the pandemic. The variance in hospitalisation length was caused by the strain on health during the pandemic peaks, which might bias the result. Statistical significance was established at $p < 0.05$. All reported *p*-values are two-tailed. We used the odds ratio (OR) and 95% confidence intervals (95% CI) to quantify the strength of the associations between covariates and dependent variables. We used SPSS version 20 (IBM Corp., Armonk, NY, USA) for statistical analysis.

*2.5. Ethics*

This study was carried out in accordance with the Code of Ethics of the World Medical Association (Declaration of Helsinki) for experiments involving humans and was approved by the local Ethical Committee of the University Hospital in Bratislava. Written informed consent for participation was obtained from all participants before enrolment in the study. No administrative permission to access the raw data used in this study was required by local authorities or the University Hospital. The raw data were fully anonymised before use. We have preserved the full anonymity of all participants.

**3. Results**

A total of 554 patients (330 men and 224 women) met the inclusion criteria and were included in the study. Overall, 330 (63.0%) patients had positive IgG or IgM anti-S SARS-CoV-2 antibodies at the time of hospital admission. IgM antibodies were positive in 326 patients (58.8%), and IgG antibodies were positive in 244 patients (46.6%). Only six IgG-positive patients had negative IgM antibodies. A significantly higher proportion of the patients with negative anti-S SARS-CoV-2 antibodies died during the hospital stay and during the follow-up (Table 1). The baseline characteristics of the cohorts of antibody-positive and antibody-negative patients are provided in Table 1. The severity of the disease categories in the cohorts, according to the NIH, is provided in Table 2. Patients with negative anti-S SARS-CoV-2 IgM and IgG antibodies at the time of admission were referred to the hospital sooner, had lower CRP and D-dimer concentrations, and were hospitalised longer. They were also more likely to be admitted to an intensive care unit and more often required invasive mechanical ventilation (Table 1). In the multivariate analysis, the negative anti-S SARS-CoV-2 IgM and IgG antibodies were associated with a higher risk of in-hospital death, independent of their age, time from symptom onset to admission, CRP, D-dimer, number of comorbidities, disease severity and baricitinib therapy (Table 3). In the Cox regression model, the negative anti-SARS-CoV-2 antibodies at the time of admission were associated with a higher risk of death during the follow-up, independent of age, time from symptom onset to admission, CRP, D-Dimer, number of comorbidities, disease severity, and baricitinib therapy (Table 4, Figure 2). The results of the analyses are provided in the supplementary material (Supplementary Materials, Tables S1–S15) in more detail.

**Table 1.** Baseline characteristics of the cohort of patients with positive anti-S-SARS-CoV-2 immunoglobulins and the cohort of patients with negative anti-S-SARS-CoV-2.

| | | Anti-SARS-CoV-2 Ig Positive *n* = 330 | Anti-SARS-CoV-2 Ig Negative *n* = 224 | *p*-Value |
|---|---|---|---|---|
| Age (years) | | 60.49 ± 13.71 | 61.11 ± 14.03 | 0.607 |
| BMI (kg/m$^2$) | | 30 (26–33) | 30 (27–35) | 0.218 |
| CRP (mg/L) | | 106 (48–161) | 82 (40–128) | 0.018 |
| IL-6 (pg/mL) | | 58 (27–101) | 56 (27–87) | 1.000 |
| N/L ratio | | 6 (4–10) | 5 (3–9) | 0.071 |
| GFR (mL/min) | | 64 (48–75) | 60 (44–71) | 0.768 |
| D-dimer (mg/L) | | 1.00 (1.00–3.00) | 1.00 (0.00–1.00) | 0.011 |
| Duration of symptoms to admission (days) | | 9 (6–11) | 7 (4–9) | 0.0001 |
| Hospitalisation length (days) | | 9 (6–14) | 11 (7–5) | 0.01 |
| Length of follow-up (months) | | 21 (20–25) | 21 (20–23) | 0.388 |
| Number of comorbidities * | | 1 (0, 2) | 1 (0, 2) | 0.566 |
| Survival in deceased patients (Days) | | 59 (14–412) | 160 (24–312) | 0.704 |
| **Categorical variables** | **Yes** / **No** | **anti-SARS-CoV-2 Ig positive, *n*/*n*-total (%)** | **anti-SARS-CoV-2 Ig negative, *n* (%)** | ***p*-value (chi-square)** |
| Death during hospitalisation | Yes | 28/330 (8.5%) | 50/224 (22.3%) | 0.0001 |
| | No | 302/330 (91.5%) | 174/224 (77.7%) | |
| Death after discharge | Yes | 12/330 (3.6%) | 16/224 (7.1%) | 0.030 |
| | No | 318/330 (96.4) | 208/224 (92.9%) | |
| Mechanical ventilation | Yes | 13/330 (3.9%) | 23/224 (10.3%) | 0.004 |
| | No | 317/330 (96.1%) | 201/224 (89.7%) | |
| Intensive care unit | Yes | 64/330 (19.4%) | 60/224 (26.8%) | 0.048 |
| | No | 226/330 (68.5%) | 164/224 (73.2%) | |
| Male gender | Yes | 197/330 (59.7%) | 133/224 (59.4%) | 0.930 |
| | No | 133/330 (40.3%) | 91/224 (40.4%) | |
| Diabetes mellitus | Yes | 62/330 (18.8%) | 39/224 (17.4%) | 0.737 |
| | No | 268/330 (81.2%) | 185/224 (82.6%) | |
| Arterial hypertension | Yes | 178/330 (53.9%) | 77/224 (54.9%) | 0.862 |
| | No | 152/330 (46.1%) | 147/224 (65.6%) | |
| MI history | Yes | 26/330 (7.9%) | 21/224 (9.4%) | 0.539 |
| | No | 304/330 (92.1%) | 203/224 (90.6%) | |
| Stroke history | Yes | 11/330 (3.3%) | 14/224 (6.3%) | 0.143 |
| | No | 319/330 (97.7%) | 210/224 (93.7%) | |
| Atrial fibrillation | Yes | 19/330 (5.8%) | 18/224 (8.1%) | 0.304 |
| | No | 311/330 (94.2%) | 206/224 (91.9%) | |
| Asthma | Yes | 26/330 (7.9%) | 22/224 (9.8%) | 0.445 |
| | No | 304/330 (92.1%) | 202/224 (90.2%) | |
| COPD | Yes | 8/330 (2.4%) | 11/224 (4.9%) | 0.153 |
| | No | 322/330 (97.6%) | 213/224 (95.1%) | |

**Table 1.** *Cont.*

|  |  | Anti-SARS-CoV-2 Ig Positive *n* = 330 | Anti-SARS-CoV-2 Ig Negative *n* = 224 | *p*-Value |
|---|---|---|---|---|
| CKD G4/G5 | Yes | 22/330 (6.7%) | 13/224 (5.7%) | 0.701 |
|  | No | 308/330 (93.3%) | 209/224 (93.3%) |  |
| Baricitinib | Yes | 97/330 (29.4%) | 92/224 (41.1%) | 0.003 |
|  | No | 295/330 (89.4%) | 132/224 (58.9%) |  |

BMI: body mass index; CRP: C-reactive protein; COPD: chronic obstructive pulmonary disease; CKD G4/G5: Chronic kidney disease G4 or G5 by KDIGO classification, GFR: glomerular filtration rate; IL-6: interleukin 6; MI: myocardial infarction; *n*: number; N/L ratio: neutrophil-to-lymphocyte ratio; *p*-value: probability; $p < 0.05$ regarded as statistically significant. The age is provided as the mean $\pm$ standard deviation (age is normally distributed). The quantitative variables except age are provided as medians (25th percentile to 75th percentile) because of a lack of normal distribution. Categorical variables are provided as the number of subjects (% of total subjects). The medians of quantitative subjects are compared using Mann–Whitney testing. The distributions of the categorical variables were compared using chi-square testing. * Number of comorbidities—the evaluated comorbidities were diabetes mellitus, arterial hypertension, myocardial infarction, stroke, atrial fibrillation, asthma, chronic obstructive pulmonary disease, chronic kidney disease grade G4 or higher according to CKD/EPI classification.

**Table 2.** Baseline COVID-19 severity of the cohort of patients with positive anti-S-SARS-CoV-2 immunoglobulins and the cohort of patients with negative anti-S-SARS-CoV-2.

| COVID Severity [†] | | |
|---|---|---|
|  | Anti-SARS-CoV-2 Ig Positive, *n*/*n*-Total (%) | Anti-SARS-CoV-2 Ig Negative, *n*/*n*-Total (%) |
| Moderate illness | 52/330 (15.8%) | 32/224 (14.3%) |
| Severe illness | 268/330 (81.2%) | 174/224 (77.7%) |
| Critical illness | 10/330 (3%) | 18/224 (8%) |
|  | 0.030 | |

[†] COVID-19 severity defined according to the National Institute of Health guidelines [15]; $p < 0.05$ regarded as statistically significant.

**Table 3.** Binary logistic regression analysis of the association of in-hospital death with anti-S SARS-CoV-2 negativity and other variables.

|  | *p*-Value | OR | 95% CI |
|---|---|---|---|
| Anti-S SARS-CoV-2 negative | 0.0001 | 3.800 | 1.844–7.829 |
| Critical disease | 0.0001 | 7.460 | 2.475–22.222 |
| Number of comorbidities (*n*) | 0.002 | 1.664 | 1.197–2.312 |
| Age (years) | 0.001 | 1.063 | 1.026–1.101 |
| D-dimer (mg/L) | 0.043 | 1.099 | 1.003–1.204 |
| **Binary regression model for separate anti-SARS-CoV-2 IgM** | | | |
|  | *p*-value | OR | 95% CI |
| Anti-S SARS-CoV-2 IgM negative | 0.0001 | 4.542 | 2.126–9.706 |
| Critical disease | 0.001 | 6.757 | 2.257–20.000 |
| Number of comorbidities (*n*) | 0.003 | 1.649 | 1.179–2.305 |
| Age (years) | 0.002 | 1.056 | 1.020–1.094 |
| D-dimer (mg/L) | 0.051 | 1.096 | 1.000–1.202 |
| N/L ratio | 0.045 | 1.043 | 1.001–1.086 |
| **Binary regression model for separate anti-SARS-CoV-2 IgG** | | | |
|  | *p*-value | OR | 95% CI |
| Anti-S SARS-CoV-2 IgG negative | 0.005 | 2.983 | 1.393–6.388 |
| Critical disease | 0.0001 | 9.434 | 3.165–27.778 |
| Number of comorbidities (*n*) | 0.042 | 1.099 | 1.003–1.203 |
| Age (years) | 0.0001 | 1.067 | 1.031–1.105 |
| D-dimer (mg/L) | 0.003 | 1.657 | 1.188–2.310 |

BMI: body mass index; CI: confidence interval; CRP: C-reactive protein; OR: odds ratio; *p*-value: probability. The duration of symptoms, baricitinib treatment, neutrophil-to-lymphocyte ratio, body mass index, CRP concentration, and IL-6 concentration were included in the binary regression analysis; however, these were not included in the model because of being insignificantly associated with death during hospitalisation or being redundant. The analyses using these variables are listed in the supplemental material (Supplementary Tables S2–S7); the collinearity analysis of the variables is provided in Supplementary Table S14; $p < 0.05$ regarded as statistically significant.

**Table 4.** Cox regression analysis of the association between death (during follow-up in patients discharged alive) and anti-S SARS-CoV-2 negativity and other variables.

| | *p*-Value | OR | 95% CI |
|---|---|---|---|
| Anti-S SARS-CoV-2 negative | 0.030 | 2.863 | 1.110–7.386 |
| Age (years) | 0.024 | 1.048 | 1.006–1.092 |
| **Cox regression model for separate anti-SARS-CoV-2 IgM** | | | |
| | *p*-value | OR | 95% CI |
| Anti-S SARS-CoV-2 IgM negative | 0.038 | 2.728 | 1.057–7.039 |
| Age (years) | 0.028 | 1.046 | 1.005–1.089 |
| **Cox regression model for separate anti-SARS-CoV-2 IgG** | | | |
| Anti-S SARS-CoV-2 IgM negative | 0.033 | 3.369 | 1.105–10.270 |
| Age (years) | 0.016 | 1.052 | 1.009–1.097 |

OR: odds ratio; *p*-value: probability. Duration of symptoms was defined as days from onset of symptoms to admission. The duration of symptoms, symptom severity, baricitinib treatment, neutrophil-to-lymphocyte ratio, body mass index, CRP concentration, D-dimer concentration, and IL-6 concentration were included in the binary regression analysis; however, they were not included in the model because of being insignificantly associated with death during hospitalisation or being redundant. The analyses with these variables included are listed in the supplemental material (Supplementary Tables S8–S13); the collinearity analysis of the variables is provided in Supplementary Table S15; $p < 0.05$ regarded as statistically significant.

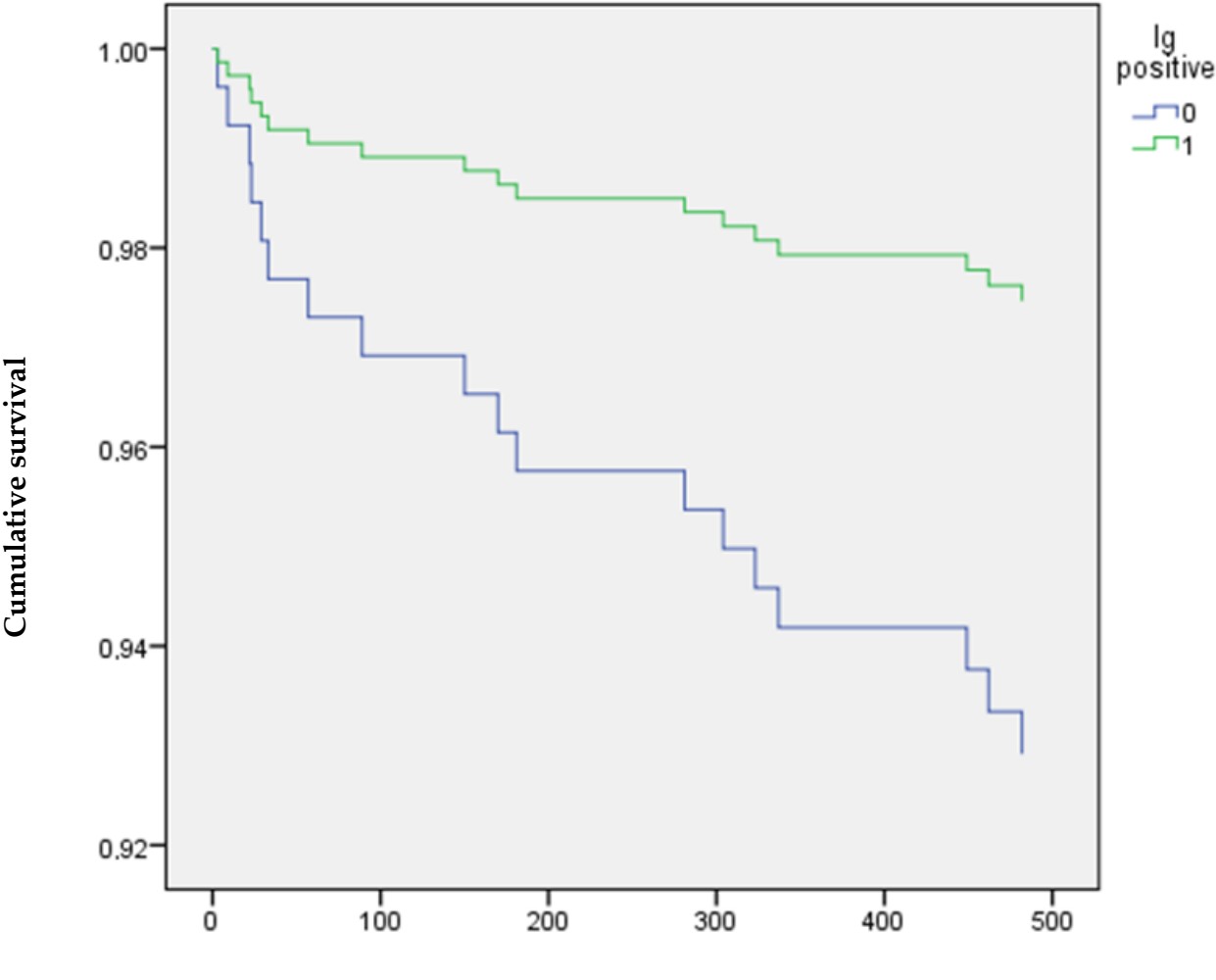

**Figure 2.** Cox regression of the association between anti-SARS-CoV-2 antibody status at the time of admission and survival after hospital discharge. Adjusted for age. Blue line: anti-SARS-CoV-2 immunoglobulin (Ig) negative; green line: anti-SARS-CoV-2 immunoglobulin (Ig) positive.

## 4. Discussion

This study has confirmed that the absence of detectable anti-S SARS-CoV-2 IgM and IgG antibody production at the time of hospital admission is associated with an increased risk of in-hospital death and all-cause death after hospital discharge in unvaccinated naive COVID-19 patients. This association was irrespective of age, number of comorbidities, the time from disease onset to antibody assessment, and other possible confounders. These findings contribute to the evidence that the development of anti-S SARS-CoV-2 antibodies counteracts the pathomechanisms precipitating the progression of tissue injury and the development of more severe disease. Our study also suggests that the effect on prognosis lasts for a long time after hospital discharge. The value of our study is further highlighted by the fact that it was conducted on the unvaccinated and immunologically naive population, and thus the results were not confounded by any pre-existing anty-SARS-CoV-2 immunity. Similar results will be difficult to obtain in the future because of mass vaccination campaigns and the high prevalence of natural immunity.

There is a large body of evidence connecting more severe disease to more vigorous antibody response in COVID-19. Several studies have found a positive association between antibody titres and disease severity and the probability of the risk of death in patients hospitalised with COVID-19. The authors of these studies have proposed that more severe disease stimulates more intensive antibody production [8–11]. This association was also apparent while measuring neutralising activity [9,11]. On the other hand, there is evidence suggesting that the prompt development of antibodies is associated with more favourable outcomes [5–7].

The main difference between the studies that have connected higher titres of anti-SARS-CoV-2 antibodies with severe and fatal disease [8–11] and later studies that identified the protective role of the early development of strong antibody response [5–7] is the temporal factor. The studies that identified the early antibody response as a protective factor assessed it at the early stages of the disease. On the other hand, authors that found an association between strong antibody response and more serious disease measured them later in the course of the disease or after its resolution.

De Vito et al. [5] found that a lower antibody titre against SARS-CoV-2 in the early stage of the disease is associated with a higher risk of in-hospital death from COVID-19, irrespective of age. However, the number of participants in their study was relatively small, and the follow-up was only 40 days. They did not measure neutralizing activity. Zohar et al. [6] found a delayed IgG1 anti-S response in COVID-19 nonsurvivors and survivors of moderate disease compared with survivors of severe disease. They also found it compromised Fcγ receptor binding and Fc effector activity in the nonsurvivors and proposed that this impairment is associated with a deficient humoral immune response. However, they found no association between anti-S IgM antibody titres and disease severity during the first week of the disease. During the second week of the disease, they found higher IgG and IgM titres in the survivors of severe disease compared with the survivors of moderate disease and the nonsurvivors. They found no difference in neutralising activity or in the titres of anti-N antibodies across the groups. The time of follow-up was 30 days. Dispinseri et al. [7] found that the presence of neutralising anti-S antibodies within the first weeks of the disease is associated with earlier virus clearance and a higher probability of survival. They suggested that a compromised immune response is a major trait of patients with severe COVID-19 [7]. Atyeo et al. [17] also emphasised the role of the anti-S antibody response as a factor shifting the disease trajectory towards a milder course. Although anti-SARS-CoV-2 IgG titres in survivors were not greater compared with deceased patients, their immune response was characterised by higher titres of anti-S IgM but decreased functional antibody responses to the nucleocapsid antigen. They found no difference in neutralising activity between the deceased and convalescent patients. They measured the antibody response approximately nine days after the onset of symptoms.

According to our results and the evidence of the studies cited above, we suggest that a delayed antibody response is more likely to occur in the early acute phase of severe or

critically ill COVID-19 patients and is associated with poor outcomes. However, later in the disease course of severe or critical illness, the production of antibodies rises significantly and eventually overtakes the production of antibodies in patients with less severe disease. Higher anti-SARS-CoV-2 antibody production is also associated with the more severe disease after disease resolution. Yan et al. [18] found that survivors of severe disease had higher titres of anti-SARS-CoV-2 antibodies one year after the resolution of an acute infection. Hansen et al. [19] also found a positive correlation between disease severity and antibody titres in convalescent individuals. Therefore, the anti-S SARS-CoV-2 antibody response in severe cases develops later; however, the results presented higher antibody titres in the convalescent stage.

Our study also found that patients with negative anti-S SARS-CoV-2 antibodies at the time of hospital admission have a higher risk of all-cause death after hospital discharge. The novelty of our study lies in the relatively long median follow-up length. Our results suggest that the effects of delayed antibody response in COVID-19 last for a long period after hospital discharge. The endpoint in our study was the all-cause death, and we were unable to analyse the specific causes of death in the deceased patients due to the lack of autopsies conducted on deceased patients during the pandemic. The exact mechanism that leads to an impaired prognosis (lasting for months) after hospital discharge in patients with delayed antibody response needs to be clarified by further studies. We propose that the more severe organ damage and its sequelae caused by more severe disease in antibody-negative subjects might contribute to the increased risk of all-cause death after hospital discharge. However, there might be other mechanisms because we were unable to analyse the precise causes of death in the deceased patients. García-Abellán et al. [20] found weaker antibody responses in patients suffering from persistence after the resolution of acute COVID-19 and suggested that antibody-mediated immune reaction plays an important role in the recovery phase.

There is substantial evidence that deregulated and compromised immune responses play a pivotal role in the pathogenesis of severe disease in COVID-19 [21]. Severe disease is associated with reduced CD3+, CD4+ CD8+, and natural killer cells [22], a higher neutrophil to lymphocyte (N/L) ratio, and higher concentrations of proinflammatory cytokines [15]. The presence of more severe disease is also associated with a lower total concentration of immunoglobulins in the acute stage of the disease [21,22].

Anti-S antibodies play a crucial role in the ability of the host to clear SARS-CoV-2 and to recover from infection. Virus-neutralising anti-S antibodies bind to the receptor-binding domain and prevent interaction with angiotensin-converting enzyme 2 (ACE2), inhibiting virus entry [23]. Blocking the interaction between SARS-CoV-2 and ACE2 in tissues not only prevents the infection of cells, but also counteracts the inhibitory effects of viral S-protein on ACE2 activity. The binding of SARS-Co-2 virions to ACE2 significantly decreases its activity in tissues, which contributes to the stimulation of the inflammatory response and probably plays a crucial role in cytokine storm development and progression [24,25]. Engineered ACE2 with augmented binding affinities for S-protein acting as decoys prevents the development of lung injury in a mouse model [26]. Monoclonal antibodies that bind to the receptor binding domain of the SARS-CoV-2 S-protein have the ability to prevent the development of severe disease if administered at the early stage of the infection [14]. Neutralizing anti-S antibodies are also regarded as being responsible for the protection against severe disease and lung and other organ injuries in vaccinees and convalescent patients and are not fully diminished by the occurrence of new variants and subvariants [27,28]. The fact that more severe disease eventually leads to higher antibody titres is believed to be caused by higher viraemia and a much stronger antigen-driven extrafollicular response [27]. The production of immunoglobulins in COVID-19 usually starts within the first or second week of infection and peaks at two months [29].

Patients with positive anti-S SARS-CoV-2 antibodies at the time of hospital admission presented with higher CRP and D-dimer concentrations. However, the CRP concentration was not associated, and the D-dimer concertation was positively associated with a higher

risk of death during hospitalisation but not during the follow-up in multivariate analyses. CRP and D-dimer are well-known prognostic markers of higher in-hospital mortality regarding COVID-19 [3,4]. Latifi-Pupovci et al. [30] described the positive correlation between CRP and D-dimer concentrations with antiSARS-CoV-2 IgG in the acute stage of the disease. To the best of our knowledge, the nature of these associations has not been elucidated yet.

This study has several limitations. We only used qualitative methods for antibody detection. Therefore, we were unable to assess the association between antibody titres with the prognosis of the disease. We used point-of-care lateral flow serological tests for quantitative analysis because they represent a more available, faster, less expensive, and less elaborate method compared with enzyme-linked immunoassays, making them a better alternative in an environment with strained human and financial resources during the COVID-19 pandemic. The point-of-care tests used in our study are characterised by a sensitivity of > 98% and a specificity of 100% relative to the enzyme-linked immunoassays [31,32]. Additionally, due to the observational character of our study and the recruitment of consecutive patients, the patient cohorts are not methodically matched for age and comorbidities. However, we do not suggest that this led to a significant bias because there is not a statistically significant difference in the medians of age and number of comorbidities and their distribution between the cohorts. We also mitigated the effects of possible confounders by using multivariate analysis. We also performed a collinearity analysis to assess if the variables in the regression models are truly independent. The results are provided in the supplementary material. The time from disease onset to admission was significantly shorter in patients with negative anti-S SARS-CoV-2 antibodies, but these associations were not present in the multivariate models. We only assessed the anti-S, not ani-N, antibodies; therefore, we are unable to conclude if the dynamic of anti-N response is also associated with a higher risk of in-hospital death and death after hospital discharge. The previous study by Zohar et al. [6] found a delayed anti-S response but no anti-N response in COVID-19 nonsurvivors. Additionally, there is substantial evidence that anti-S antibodies are crucial in the recovery from COVID-19 [23,25,26,28]. There is even evidence that anti-N antibodies might contribute to the pathogenesis of the cytokine storm in COVID-19 [33]. Therefore, we focused on anti-S antibody response and utilised just the test to assess the presence of anti-S antibodies. The follow-up period was not uniform for all patients; however, there was no difference in the median length of the follow-up between the cohorts; therefore, we suggest that this fact did not create a bias in the results. The baricitinib treatment was utilised in patients admitted from March 2021 according to the institutional guidelines. The institutional board decided to add baricitinib to the institutional guidelines according to the results of a study by Rodriguez-Garcia et al. [34]. There was no difference in baricitinib utilisation between the cohorts, and baricitinib treatment was included in the multivariate analysis; therefore, we suggest that baricitinib treatment did not create a bias in the results. The causes of death in the patients that died during the hospitalisation were assessed by autopsies. However, the autopsy is also not an absolutely exact method to determine the cause of death regarding COVID-19 [35].

## 5. Conclusions

Negative anti-S SARS-CoV-2 IgM and IgG at the time of hospital admission are associated with a higher risk of in-hospital death and also with a higher risk of all-cause death after hospital discharge. Our findings support the theory that a delayed anti-S antibody response is associated with disease severity in COVID-19, which might impair the prognosis of patients also after hospital discharge. However, further studies are needed to clarify the role of the delayed antibody response in the pathogenesis of COVID-19 and its complications.

**Supplementary Materials:** The following supporting information can be downloaded at: https://www.mdpi.com/article/10.3390/idr14060100/s1, Table S1: Kolmogorov-Smirnov test for normal distribution of quantitative variables obtained at the time of admission in the characteristics of the cohort of patients with positive anti-S-SARS-CoV-2 immunoglobulins and cohort of patients with negative anti-S-SARS-CoV-2 immunoglobulins; Table S2: Binary logistic regression model—forward stepwise method of association of in-hospital death with anti-S SARS-CoV-2 negativity and other variables. Variables in the equation of regression model; Table S3: Binary logistic regression model—forward stepwise method of association of in-hospital death with anti-S SARS-CoV-2 negativity and other variables. Variables not in the equation of regression model; Table S4: Binary logistic regression model—forward stepwise method of association of of in-hospital death with anti-S SARS-CoV-2 IgM negativity and other variables. Variables in the equation of regression model; Table S5: Binary logistic regression model—forward stepwise method of association of in-hospital death with anti-S SARS-CoV-2 IgM negativity and other variables. Variables not in the equation of regression model; Table S6: Binary logistic regression model—forward stepwise method of association of of in-hospital death with anti-S SARS-CoV-2 IgG negativity and other variables. Variables in the equation of regression model; Table S7: Binary logistic regression model—forward stepwise method of association of in-hospital death with anti-S SARS-CoV-2 IgG negativity and other variables. Variables not in the equation of regression model; Table S8: Cox regression model—forward stepwise method of association of in-hospital death with anti-S SARS-CoV-2 negativity and other variables. Variables in the equation of regression model; Table S9: Cox regression model—forward stepwise method of association of in-hospital death with anti-S SARS-CoV-2 negativity and other variables. Variables not in the equation of regression model; Table S10: Cox regression model—forward stepwise method of association of in-hospital death with anti-S SARS-CoV-2 IgM negativity and other variables. Variables in the equation of regression model; Table S11: Cox regression model—forward stepwise method of association of in-hospital death with anti-S SARS-CoV-2 IgM negativity and other variables. Variables not in the equation of regression model; Table S12: Cox regression model—forward stepwise method of association of in-hospital death with anti-S SARS-CoV-2 IgG negativity and other variables. Variables in the equation of regression model; Table S13: Cox regression model—forward stepwise method of association of in-hospital death with anti-S SARS-CoV-2 IgG negativity and other variables. Variables not in the equation of regression model; Table S14: Collinearity analysis of variables in the association with in-hospital death; Table S15: Collinearity analysis of variables in association with death during the follow-up.

**Author Contributions:** J.J. drafted the manuscript, co-developed the study design, and participated in the data collection, analysis, and interpretation of the results. A.N. participated in the data collection. M.D. co-drafted the manuscript and participated in the data analysis and interpretation of the results. P.M. participated in the data analysis and interpretation of the results. I.S. participated in the interpretation of results, revised the manuscript, and supervised the study. V.B. and M.K. participated in the data collection. M.P. participated in the data collection. M.Č. participated in the data analysis and created figures. P.S. drafted the manuscript, co-developed the study design, and participated in the analysis and interpretation of the results. All authors drafted the manuscript for important intellectual content and read and approved the final manuscript. All authors have read and agreed to the published version of the manuscript.

**Funding:** This research received no external funding.

**Institutional Review Board Statement:** The study was conducted in accordance with the Declaration of Helsinki, and approved by the Institutional Ethics Committee of University Hospital in Bratislava (protocol code: KR022020, 30 March 2020).

**Informed Consent Statement:** Informed consent was obtained from all subjects involved in the study.

**Data Availability Statement:** The data presented in this study are available on request from the corresponding author.

**Acknowledgments:** The authors would like to thank Tomáš Vinař, associate professor in the Department of Applied Informatics, Faculty of Mathematics, Physics and Informatics, Comenius University in Bratislava, Slovakia.

**Conflicts of Interest:** The authors declare no conflict of interest.

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
