# Peer review of "Anti-SARS-CoV-2 Antibody Status at the Time of Hospital Admission and the Prognosis of Patients with COVID-19: A Prospective Observational Study"

_2036-7449, doi:10.3390/idr14060100_

Round 1

Reviewer 1 Report

The authors seek to add a time-dependent aspect to the study of antibody response to SARS-CoV-2 by examining data associated with testing at the time of admission to the hospital. The authors should consider reading the manuscript ``Prevalence Estimation and Optimal Classification Methods to Account for Time Dependence in Antibody Levels" by Bedekar et. al. where this phenomenon has been examined mathematically and this work should be commented on and referred to in the manuscript. Also the manuscript makes reference to quantitative variables in the cohort being expressed as medians and failing to be normally distributed - this should be made far more clear, qualified and perhaps quantified. Also, there are alternative tests and this would make an interesting comparison and give the reader more context. The employed multivariate binary logistic regression model should be explicitly stated somewhere and the specific forward stepwise procedure selected should be both prescribed and motivated. Again, it would be convincing and helpful to the reader to be shown a variety of techniques especially if the variation in the results is small. Also, the manuscript has spelling errors (e.g. "Lenght") and there are too many uses of undefined and unclear expressions (e.g. "worse overall survival"). These relative modifiers should be replaced with specific, clean clear definitions and classifications. Finally, the paper could be shortened, the data and the thesis of the paper is interesting but statistical analysis and specification of methods used needs to be added - much of the writing could be shortened by putting in tables and simply listing numerical values when appropriate.  

Author Response

Dear reviewer. 

We are very gratefull for your valuable coments and suggestions. We tried to do our best to revise the manuscript according to your suggestions. Below, we are providing the list of revisions as a response to your commenst:

  • The authors should consider reading the manuscript ``Prevalence Estimation and Optimal Classification Methods to Account for Time Dependence in Antibody Levels" by Bedekar et. al. where this phenomenon has been examined mathematically and this work should be commented on and referred to in the manuscript.
    • We cited this preprint in discusion section further clarifiing antibody dinamics in COVID-19. 
  • Also the manuscript makes reference to quantitative variables in the cohort being expressed as medians and failing to be normally distributed - this should be made far more clear, qualified and perhaps quantified. Also, there are alternative tests and this would make an interesting comparison and give the reader more context.
    • In revised manuscript, we performed Kolmogorov-Smirnov test to assess the normal distribution and provided the resuslt of KS test in Supplementary material (Supplemetanry table S1) to provide more insight to the reader as you suggested.
  • The employed multivariate binary logistic regression model should be explicitly stated somewhere and the specific forward stepwise procedure selected should be both prescribed and motivated. Again, it would be convincing and helpful to the reader to be shown a variety of techniques especially if the variation in the results is small.
    • In revised manuscript, we performed forward stepwise procedure in binary logistic regression analysis and also Cox regression analysis and clearly name all variables that were included in to the analysis at the baseline and also in final regression models. We also provided the raders with alss steps in forward stepwise procedure in supplementarz material.
  • Also, the manuscript has spelling errors (e.g. "Lenght") and there are too many uses of undefined and unclear expressions (e.g. "worse overall survival"). These relative modifiers should be replaced with specific, clean clear definitions and classifications. Finally, the paper could be shortened, the data and the thesis of the paper is interesting but statistical analysis and specification of methods used needs to be added - much of the writing could be shortened by putting in tables and simply listing numerical values when appropriate.  
    • We corrected typos and unclear expressions with more precise extpressions like "higher risk of death during follow-up" and we tried to shorten the "Results" section as you suggested. However we were unable to shorten other parts because we had to fulfill the recommendations of other revirwers. 

Kind regards

Peter Sabaka, corresponding author

Reviewer 2 Report

This study further confirms and builds on previous reports of an association of covid-19 antibody responses (or lack thereof) and  COVID-19 disease severity and death. This group reports they are the first to use the metric of antibody at time of admission and long term mortality (all cause).

The results are presented well and the manuscript is easy to follow. 

Major comments: 

- It's important to distinguish antibody titres and neutralizing antibodies. They do not necessarily correlate and I felt the references included in both the introduction and the discussion sections describe a mix of papers using antibody titres and some measuring neutralization. Ensure that you make this important distinction

- Highlighting that this work was done in an unvaccinated population at the time prior to Omicron (and mass infections) is valuable because its natural infection data without the confounders of immunity induced by vaccine or prior infection.  The novelty of the study increases if this is highlighted as these studies are nearly impossible to do now. 

- The univariate analysis shown in table 1 compared the antibody positive and antibody negative groups, whereas the regression analysis compares survivors and non survivors.  Particularly for the regression data presented in table 2,  I don't know which variable were included in the model, only those with p<0.05 as stated in the methods? Either make another table, similar to table 1, showing the univariate analysis of the variable between survivors and non-survivors or find some way to present which ones were included in the regression. 

- Also explain what variable were incorporated into the Cox regression vs what was presented in table 3. 

- The study found that although those with no early antibody response at the time of admission were more likely to have severe covid complications and higher death outcomes, they presented with lower CP and D-dimer concentrations. This contradicts other papers showing these markers of inflammation are known to be predictors of disease severity (as noted by the authors in the introductions) but this result is not discussed at all and it needs to be. 

Minor comments: 

- Add some more recent literature and reviews to the background references on Ab levels and severity ( H. Qi et al, Nat Immunology review,2022) , even include some that discuss vaccine protection against disease severity ( M. Davenport, Nat Med 2021)

- Why test both IgG and IgM, what do they both indicate?

- The group names in Table 1 are incorrect, they both say Ig positive.

- were all variables monitored included in Table 1? If so, what about smoking, Renal Disease/ Dialysis, and Cardiovascular disease. Those are obvious comorbidities that could obscure the findings

- In the stats section of the methods, why do you use log regression for in-hospital deaths, and cox regression for the death after discharge. Simply state why the difference.

- In the discussion there is mention that the antibody assay used has high sensitivity and specificity. What are they, and is there a reference for this assay? Please include 

Author Response

We are very grateful for your valuable comments and suggestions. We tried to do our best to revise the manuscript according to your suggestions. Below, we are providing the list of revisions as a response to your comments:

Major comments: 

  • It's important to distinguish antibody titres and neutralizing antibodies. They do not necessarily correlate and I felt the references included in both the introduction and the discussion sections describe a mix of papers using antibody titres and some measuring neutralization. Ensure that you make this important distinction
    • In Introduction and in the Discussion, we specifically provided the information if the cited paper refers to neutralising antibodies or titre of antibodies without the neutralisation test performed. 
  • Highlighting that this work was done in an unvaccinated population at the time prior to Omicron (and mass infections) is valuable because its natural infection data without the confounders of immunity induced by vaccine or prior infection.  The novelty of the study increases if this is highlighted as these studies are nearly impossible to do now. 
    • In the Discussion section, we highlighted this fact by: "The value of our study is further highlighted by the fact, that it was conducted on the unvaccinated and immunologically naïve population and thus the results were not confounded by any preexisting anty-SARS-CoV-2 immunity. Similar results will be difficult to obtain in the future because of mass vaccination campaigns and the high prevalence of natural immunity. "
  • The univariate analysis shown in table 1 compared the antibody positive and antibody negative groups, whereas the regression analysis compares survivors and non survivors.  Particularly for the regression data presented in table 2,  I don't know which variable were included in the model, only those with p<0.05 as stated in the methods? Either make another table, similar to table 1, showing the univariate analysis of the variable between survivors and non-survivors or find some way to present which ones were included in the regression. 
    • In the revised manuscript, we performed a forward stepwise procedure in binary logistic regression analysis and also Cox regression analysis and clearly name all variables that were included in the analysis at the baseline and also in final regression models. We also provided the readers with all steps in the forward stepwise procedure in the supplementary material.
  • Also explain what variable were incorporated into the Cox regression vs what was presented in table 3. 
    • We provided the list of variables in the footnote of the table with Cox regression analysis: "The duration of symptoms, baricitinib treatment, neutrophil-to-lymphocyte ratio, body mass index, CRP concentration, and IL-6 concentration were included in binary regression analysis, however, were not included into the model because being insignificantly associated with death during hospitalisation or redundant. The analyses with these variables included are listed in the supplementary material."
  • The study found that although those with no early antibody response at the time of admission were more likely to have severe covid complications and higher death outcomes, they presented with lower CP and D-dimer concentrations. This contradicts other papers showing these markers of inflammation are known to be predictors of disease severity (as noted by the authors in the introductions) but this result is not discussed at all and it needs to be. 
    • We addressed this issue in Discussion " 

      Patients with positive anti-S SARS- CoV-2 antibodies at the time of hospital admission presented with higher CRP and D-dimer concentrations. However, the CRP concentration was not associated, and the D-dimer concertation was positively associated with a higher risk of death during hospitalization, but not during the follow-up in multivariate analyses. CRP and D–dimer are well-known prognostic markers of higher in-hospital mortality in COVID-19.3,4 Latifi-Pupovci at al. 30 described a positive correlation of CRP and D-dimer concentrations with antiSARS-CoV-2 IgG in the acute stage of the disease. To our knowledge, the nature of these associations has not been elucidated yet."

Minor comments: 

  • Add some more recent literature and reviews to the background references on Ab levels and severity ( H. Qi et al, Nat Immunology review,2022) , even include some that discuss vaccine protection against disease severity ( M. Davenport, Nat Med 2021)
    • We cited these papers in Introduction and provided the readers with more recent findings based on the results of these papers.
  • Why test both IgG and IgM, what do they both indicate?
    • The commercially available point-of-care test we used provides results for both IgM and IgG antibodies. We were trying to demonstrate the utilisation of these point-of-care tests in the assessment of prognosis in admitted naive patients. The resuslt for both IgG and IgM were comparable and positivity of both IgM and IgG indicates antibody response at the time of hospital admission. We provided the readers with the results of separate analyses for IgM and IgG in the supplementary material. We didn´t include this material in the main text because we find the abundance of data o overwhelming.
  • The group names in Table 1 are incorrect, they both say Ig positive.
    • We corrected this issue.
  • were all variables monitored included in Table 1? If so, what about smoking, Renal Disease/ Dialysis, and Cardiovascular disease. Those are obvious comorbidities that could obscure the findings
    • We added the history of severe kidney disease. In previous version, we assessed the prevalence of arterial hypertension and the history of myocardial infarction. We omitted the history of coronary heart disease because of the low utilisation of coronagraphy in Slovakia.
  • In the stats section of the methods, why do you use log regression for in-hospital deaths and cox regression for death after discharge. Simply state why the difference.
    • We used logistic regression instead of Cox regression to analyse in-hospital deaths because of the inconsistency of hospitalisation lengths that varied during the pandemic. The variance in hospitalisation length was caused by the high strain during the pandemic peaks which might bias the result. We also found the time-to-event factor negligible in analysing the risk of in-hospital death
  • In the discussion there is mention that the antibody assay used has high sensitivity and specificity. What are they, and is there a reference for this assay? Please include 
    • We included the specificity and sensitivity of the point-of-care test in the discussion section of the manuscripts.

Reviewer 3 Report

Dear authors,

I have received an article examining the effects of antibody response to SARS-CoV-2 S protein at the time of hospital admission. The authors found that this antibody's absence confers an increased mortality risk. While this is undoubtedly a welcomed finding, some issues need to be addressed:

Major issues:

Abstract

- Please include the odds ratio in the abstract, as the abstract should be a stand-alone finding.

Introduction

- "Delayed antibody response against severe acute respiratory syndrome coronavirus 2 (SARS-CoV-2) antigens has recently been identified as a predictor of inhospital mortality" --> How recent? Reference no. 6 was published in 2020, in the early days of the pandemic.

- There is a lack of aim of this study, despite the drive being mentioned in the abstract

- The authors need to justify their novelty in this journal. A preliminary search in Google led me to numerous articles being published in this article. Either the novelty section should be rephrased or removed entirely.

- The authors need to mention why only anti-S antibodies are being studied. Why not other antibodies?

Methods:

- The authors need to clarify the inclusion criteria better. Do patients need to be categorized as having severe COVID-19 infections? If so, what are the requirements for extreme conditions? Also, if only severe patients are analyzed, the title should reflect this.

- "a maximum of 21 days preceding admission" --> Why 21 days preceding admission? This is a severe potential bias as a positive or negative finding from 21 days before may not reflect current results upon entry.

- The authors need to describe if one patient has multiple tests of antibodies; which one is being used?

- The authors need to specify how long after the discharge the patient is being followed-up. The authors mentioned in the discussion section that the potential cause of death may be due to immunologic reasons. However, the authors need to modify that claim as there is no way of knowing whether they died due to COVID-19 reasons or not related to COVID-19 (suicide, accidents, etc.)

- "Regarding immunomodulatory therapy, all patients admitted due to severe COVID-19 received dexamethasone at a dose of 6 mg daily. Patients who required high-flow oxygen or mechanical ventilation were treated by baricitinib at a dose of 4 mg daily" --> The authors need to specify whether this is according to the local or institutional guidelines. If it is according to the local policies, the authors must give a proper citation.

- "The results were read after 15 min (max 20 min) by the naked eye" --> How many interpreters did this? Were they trained? What's the inter-rater and intra-rater observability between them? What are this machine's sensitivity, specificity, PPV, and NPV?

- Baricitinib is approved for use in 2022; this study was conducted in 2020-2021. Please explain.

- Figure 1 should be expanded in the "Died during hospitalization" part. How many of the deceased have comorbidities? How many died due to COVID-19?

- Figure 2 should be in the results section and be explained in greater detail. The authors should also acknowledge that the survival analysis graph crosses twice, making the finding insignificant.

Results

- "Of hospitalized patients, 78 (14.1%) died during the hospital stay, and the remainder were live discharges. During the hospital stay, 28 (8.5%) antibody positive and 50 (22.3%) antibody-negative patients died (p < 0.0001)." --> These two sentences repeat each other. Please simplify them.

- Please express the duration of follow-ups in months and not days to make it more intuitive.

- In table 1, the headings in the numerical section are missing. 

- The authors should analyze the difference between IgG and IgM.

- Please do not express the p-value as <0.05; please include the whole number.

- The categorical variable part is wrong; it should be a 2x2 table. Therefore, for variables death during hospitalization etc., there should be a yes and no, making it a 2x2 table.

- The authors should include the overall presence of comorbidities instead of breaking them down immediately, as the specific comorbidities may be underpowered to detect significance.

- The authors mentioned that only variables with a p-value of <0.05 will be included in the regression analysis. Therefore, why is age included?

Discussion 

- "Our study was the first to conclude that this effect on prognosis lasts for a long time after hospital discharge" --> This is too strong of a statement for this study. No conclusions can be reached from this, only some suggestions. Similarly, any "conclusions" made in the discussion section should also be removed.

- "Anti-S antibodies.....administered at the early stage of the infection" --> This should be included in the introduction section explaining why this study looks at the anti-S antibodies.

Minor issues:

- The abstract in MDPI is unstructured. Please revise this.

- There are no line numbers which are required by the journals.

- There is no need for an abbreviation list; it is only required for Springer's journals.

- Please confirm that the "Department of Infectology and Geographical Medicine" is correct and not a typo.

- There are many spelling errors, such as "positive paints" in the results section or "time form" in the discussion section. Please check the manuscript thoroughly or seek a professional English proofreader.

- The heading of table 1 is too long. The table can be simplified by removing the sentences after "time of admission".

Author Response

Abstract

  • Please include the odds ratio in the abstract, as the abstract should be a stand-alone finding.
    • We included odds ratios of the association of immunoglobulin negativity with in in-hospital death and death during follow-up to the abstract according to the results of regression models.

Introduction

  • "Delayed antibody response against severe acute respiratory syndrome coronavirus 2 (SARS-CoV-2) antigens has recently been identified as a predictor of inhospital mortality" --> How recent? Reference no. 6 was published in 2020, in the early days of the pandemic.
    • The study form 2020 is indeed not recent so we rephrased the sentence: Delayed antibody response against severe acute respiratory syndrome coronavirus 2 (SARS-CoV-2) antigens has been identified as a predictor of in-hospital mortality.
  • There is a lack of aim of this study, despite the drive being mentioned in the abstract
    • We added the aim of the study to the Introductions
  • The authors need to justify their novelty in this journal. A preliminary search in Google led me to numerous articles being published in this article. Either the novelty section should be rephrased or removed entirely.
    • We rephrased the novelty section and tied to justify the novelty of our study by comparing our results to published papers in the discussion. The novelty of our study lies in the relatively long median follow-up length. Our results suggest that the effects of delayed antibody response in COVID-19 last for a long time period from hospital discharge.
  • The authors need to mention why only anti-S antibodies are being studied. Why not other antibodies?
    • We elucidated the reasons why we focused on anti-S antibodies in discussion: ``We only assess the anti-S, not ani-N antibodies, so we are unable to conclude if the dynamic of anti-N response is also associated with a higher risk of in-hospital death and death after hospital discharge. The previous study by Zohar et al. found the delayed anti-S response but no anti-N response in COVID-19 non-survivors. Also, there is substantial evidence that anti-S antibodies are crucial in the recovery from COID-19. There is even evidence that anti-N antibodies might contribute to the pathogenesis of cytokine storm in COVID-19. 32 Therefore we focused on anti-S antibody response and utilised just the test to assess the presence of anti-S antibodies. ``

Methods:

  •  The authors need to clarify the inclusion criteria better. Do patients need to be categorized as having severe COVID-19 infections? If so, what are the requirements for extreme conditions? Also, if only severe patients are analyzed, the title should reflect this.
    • We concretize inclusion criteria in the methods section: "COVID-19 with a moderate, severe, or critical disease as a reason for hospital admission as defined in the National Institutes of Health (NIH) guidelines."
  • "a maximum of 21 days preceding admission" --> Why 21 days preceding admission? This is a severe potential bias as a positive or negative finding from 21 days before may not reflect current results upon entry.
    • We defined the 21 days as the upper interval of time from the occurrence of COVID-19 positivity by PCR test to the development of severe disease in the study project because we anticipated that strain to the healthcare system might prevent the collection of samples during admission. However, during the study realisation, we were finaly able to collect the sample for PCR at the time of admission in all patients. We left the inclusion criteria to be "21 days prior admission" by mistake. We corrected the inclusion criteria in methods and deleted the 21 days statement. 
  • The authors need to describe if one patient has multiple tests of antibodies; which one is being used?
    • If the patient had 2 or more antibody tests, only the result of the first test was included in the analysis. We added this fact into the methods.
  • The authors need to specify how long after the discharge the patient is being followed-up. The authors mentioned in the discussion section that the potential cause of death may be due to immunologic reasons. However, the authors need to modify that claim as there is no way of knowing whether they died due to COVID-19 reasons or not related to COVID-19 (suicide, accidents, etc.)
    • We addressed this issue in discussion: "The exact mechanism that leads to impaired prognosis lasting for months after hospital discharge in patients with delayed antibody response needs to be clarified by further studies. We propose that the more severe organ damage and its sequelae caused by more severe disease in antibody negative subjects might contribute to the increased risk of all-cause death after hospital discharge. However, there might be other mechanisms because we were unable to analyse precise causes of death in deceased patients. "
  • "Regarding immunomodulatory therapy, all patients admitted due to severe COVID-19 received dexamethasone at a dose of 6 mg daily. Patients who required high-flow oxygen or mechanical ventilation were treated by baricitinib at a dose of 4 mg daily" --> The authors need to specify whether this is according to the local or institutional guidelines. If it is according to the local policies, the authors must give a proper citation.
    • We administered the therapy according to institutional guidelines. We added this fact in the methods. 
  • "The results were read after 15 min (max 20 min) by the naked eye" --> How many interpreters did this? Were they trained? What's the inter-rater and intra-rater observability between them? What are this machine's sensitivity, specificity, PPV, and NPV?
    • We provided the specificity and sensitivity of the test according to 2 studies in the discussion. All readings were performed by the same person from the blood collected at the time of admission. We added this fact to the methods. 
  • Baricitinib is approved for use in 2022; this study was conducted in 2020-2021. Please explain.
    • The baricitinib was added to our institutional guidelines sooner than to NIH or IDSA gideines. We explained the soon inclusion of baricitinib in the discussion.  
  • Figure 1 should be expanded in the "Died during hospitalization" part. How many of the deceased have comorbidities? How many died due to COVID-19?
    • We provided the comorbidities and their distribution in table 2. All patients that were included in the study had COVID-19 concluded as the cause of death by autopsy/ In the Slovak republic, we had a very strict policy to define the cause of death in hospitalised SASR-CoV-2 positive patients and autopsy was mandatory.
  • Figure 2 should be in the results section and be explained in greater detail. The authors should also acknowledge that the survival analysis graph crosses twice, making the finding insignificant.
    • The Figure 2 has been moved to methods section by MDPI office. We moved it to the results. We repeated the Cox regression according to your suggestions and the new graph is without crossing.

Results

  • "Of hospitalized patients, 78 (14.1%) died during the hospital stay, and the remainder were live discharges. During the hospital stay, 28 (8.5%) antibody positive and 50 (22.3%) antibody-negative patients died (p < 0.0001)." --> These two sentences repeat each other. Please simplify them.
    • We simplified the results section and removed repeating data.
  • Please express the duration of follow-ups in months and not days to make it more intuitive.
    • We converted the days to the months. The length of follow-up in months are indeed more informative
  • In table 1, the headings in the numerical section are missing.
    • We corrected table 1.
  • The authors should analyze the difference between IgG and IgM.
    • In revised manuscript, we analysed IgM and IgG also separately. 
  • Please do not express the p-value as <0.05; please include the whole number.
    • We provided the whole p number in all tables

  • The categorical variable part is wrong; it should be a 2x2 table. Therefore, for variables death during hospitalization etc., there should be a yes and no, making it a 2x2 table.
    • We corrected the table 1 according to your suggestions.
  • The authors should include the overall presence of comorbidities instead of breaking them down immediately, as the specific comorbidities may be underpowered to detect significance.
    • We included the overal number of comorbidities in the multivariate analyses.
  • The authors mentioned that only variables with a p-value of <0.05 will be included in the regression analysis. Therefore, why is age included?
    • We recomputed binary regression analyses and cox regression analyses according to suggestions of reviewer 2 and used forward stepwise method and defind p value below 0.05 as significant for inclusion of each variable to the model.

Discussion 

  • "Our study was the first to conclude that this effect on prognosis lasts for a long time after hospital discharge" --> This is too strong of a statement for this study. No conclusions can be reached from this, only some suggestions. Similarly, any "conclusions" made in the discussion section should also be removed.
    • We reformulated all the statements in the discussion according to your suggestions.
  • "Anti-S antibodies.....administered at the early stage of the infection" --> This should be included in the introduction section explaining why this study looks at the anti-S antibodies.
    • We moved the information regarding monoclanal antibodies in to the introduction and tied to provide further clarrification why we focused on the anti-S antibodies.

Minor issues:

  • The abstract in MDPI is unstructured. Please revise this.
    • We corrected the abstract.
  • There are no line numbers which are required by the journals.
    • The lines were not included by the MDPI office.
  • There is no need for an abbreviation list; it is only required for Springer's journals.
    • We removed abbrevation list.
  • Please confirm that the "Department of Infectology and Geographical Medicine" is correct and not a typo.
    • It is the correct official name of our department in english.
  • There are many spelling errors, such as "positive paints" in the results section or "time form" in the discussion section. Please check the manuscript thoroughly or seek a professional English proofreader.
    • The revuised manuscript was proofred by profesional service and we corrected the tipos. 
  • The heading of table 1 is too long. The table can be simplified by removing the sentences after "time of admission".
    • We corrected the table names according to your suggestion.

Reviewer 4 Report

Summary:

The current publication looks at the association of the absence of anti-SARS-CoV-2 Ig levels, at the time of hospital admission, with post-hospitalization outcomes in unvaccinated patients.

Comments/questions:

If my understanding of figure 2 is correct, is it showing that anti-SARS-CoV2 immunoglobulin (Ig) positive patients had lower survival. Has the graph been mislabeled, or am I interpreting this figure wrong?

Table 1 shows anti-SARS-CoV2 immunoglobulin (Ig) positive for both column headers. This must be mislabeled. Please fix this.

In the statistical analysis section, the authors indicate that they used a step-wise regression approach to select for significant variables. 

I would also recommend that the authors evaluate the independence of their variables by looking at Variance Inflation Factor (VIF) to see which variables are correlated.

Since forward selection procedure gives you the best model using AIC or BIC, is there a reason the authors only selected the statistically significant variables, instead of the best model proposed by the selection process?

What role did disease severity play in the follow-up outcomes? I see from Table 1 that anti-SARS-CoV-2 Ig negative patients had significantly higher incidence of mechanical ventilation. Is it possible for the authors to incorporate some metric of disease severity in their analysis?

Suggestions:

1) Please review the manuscript and fix typos and mislabeled items.

2) Please provide (at least in supplementary material) the initial set of variables used in regression analysis, and the subsequent results from variable selection.

3) I would also like to see if the chosen variables were truly independent. This can be achieved through VIF analysis.

4) If possible, I would like to see a disease/symptom severity score at the time of admission, and it's association with outcomes.

Author Response

Dear reviewer

We are very grateful for your valuable comments and suggestions. We tried to do our best to revise the manuscript according to your suggestions. Below, we are providing the list of revisions as a response to your comments:

Comments/questions:

If my understanding of figure 2 is correct, is it showing that anti-SARS-CoV2 immunoglobulin (Ig) positive patients had lower survival. Has the graph been mislabeled, or am I interpreting this figure wrong?

- The table was mislabeled. Thank you very much for pointing it out. We corected the labeling in the revised manuscript.

Table 1 shows anti-SARS-CoV2 immunoglobulin (Ig) positive for both column headers. This must be mislabeled. Please fix this.

- The table was mislabeled. Thank you very much for pointing it out. We corected the labeling in the revised manuscript.

In the statistical analysis section, the authors indicate that they used a step-wise regression approach to select for significant variables. 

I would also recommend that the authors evaluate the independence of their variables by looking at Variance Inflation Factor (VIF) to see which variables are correlated.

- We used VIF analysis which found no significant colinearirty of variables. The result of VIF is provided in supplementary material.

Since forward selection procedure gives you the best model using AIC or BIC, is there a reason the authors only selected the statistically significant variables, instead of the best model proposed by the selection process?

- Unfortunately, SPSS do not alow us to use AIC or BIC in Cox regression. However, because of relative low p values of assocation of antibody presnece with deat during follow up, we believe that also by using models utilising BIC, the results will be comparable. 

What role did disease severity play in the follow-up outcomes? I see from Table 1 that anti-SARS-CoV-2 Ig negative patients had significantly higher incidence of mechanical ventilation. Is it possible for the authors to incorporate some metric of disease severity in their analysis?

- In incoroporated the metrics of dieasease severity according to NIH guidelines.

Suggestions

1) Please review the manuscript and fix typos and mislabeled items.

- We corected the labeling and typos in the revised manuscript.

2) Please provide (at least in supplementary material) the initial set of variables used in regression analysis, and the subsequent results from variable selection.

- We provided the initial set of variables in the supplementary matrial and we also provided the all steps of forward stewise regression analyses.

3) I would also like to see if the chosen variables were truly independent. This can be achieved through VIF analysis.

- We porformed the VIF analysisi and provided the results in supplementary material

4) If possible, I would like to see a disease/symptom severity score at the time of admission, and it's association with outcomes.

- We added the disease severity according to National Institute of Healt guidelines to the analyses. 

Round 2

Reviewer 3 Report

Dear authors, 

Thank you for addressing almost all the inquiries. However, there is a glaring issue in Figure 2 as it is not described anywhere in the results. Survival is higher in those with negative antibodies than in those with positive antibodies. This finding contradicts the whole regression analysis. Please do a further check on the statistics, or else please explain this anomaly.

Also, it is interesting that all dead patients underwent autopsies. Are there any results on the autopsies, or did the authors plan on writing a separate article on this? Also, autopsies are not a surefire way of determining the cause of death (COD) is due to COVID-19. I refer the authors to this article (https://doi.org/10.1186/s41935-022-00288-0), and I hope it is included in the limitation (i.e. autopsies do not definitively determine the COD is due to COVID-19)

Author Response

Dear reviewer

Thank you again for your positive review and valuable suggestions. 

  • Thank you for addressing almost all the inquiries. However, there is a glaring issue in Figure 2 as it is not described anywhere in the results. Survival is higher in those with negative antibodies than in those with positive antibodies. This finding contradicts the whole regression analysis. Please do a further check on the statistics, or else please explain this anomaly.
    • The cohorts' names in figure 2 were mislabeled (switched) by mistake. Of course, patients with negative Ig had worse survival. In the revised manuscript, we corrected the labeling in Figure 2. 
  • Also, it is interesting that all dead patients underwent autopsies. Are there any results on the autopsies, or did the authors plan on writing a separate article on this? Also, autopsies are not a surefire way of determining the cause of death (COD) is due to COVID-19. I refer the authors to this article (https://doi.org/10.1186/s41935-022-00288-0), and I hope it is included in the limitation (i.e. autopsies do not definitively determine the COD is due to COVID-19)
    • All patients deceased in hospitals with COVID-19  in Slovakia underwent autopsies.  It was a policy of the Ministry of Health to obtain exact statistics of  COVID-19 deaths. We are planning to assess these data, but unfortunately, we had no time until now. We addressed the issue of uncertainty of the cause of death assessed by the autopsies in the discussion and cited the paper you suggested. "The causes of death in patients that died during the hospitalization in o were assessed by autopsies. However, the autopsy is also not absolutely an exact method of the cause of death determination in COVID-19."

Kind regards

Peter Sabaka

Round 3

Reviewer 3 Report

All seems good now. Thank you for clarifying